# Can Blockchain Payment Services Influence Customers' Loyalty Intention in the Hospitality Industry? A Mediation Assessment

Rashed Al Karim [1,*] , Md Karim Rabiul [2] , Mahima Ishrat [1], Pornpisanu Promsivapallop [2] and Sakia Kawser [1]

[1]  School of Business Administration, East Delta University, Noman Society, Chittagong 4209, Bangladesh
[2]  Faculty of Hospitality and Tourism, Prince of Songkla University, Kathu, Phuket 83120, Thailand
*   Correspondence: alkarim.rashed@gmail.com

**Abstract:** This study analyzes the impact of blockchain mobile payment services on customer loyalty intention through the mediating role of service quality, privacy and security, and customer satisfaction in the Bangladeshi hospitality industry. Data were collected through a survey using a structured questionnaire from 326 respondents who stayed in 4- and 5-star hotels in Chattogram and Cox's Bazar. Respondents' (N = 326) opinions were analyzed employing Smart PLS software. The results ensure that privacy and security and customer satisfaction mediate the blockchain-based mobile payment services and loyalty intention relationship. However, service quality does not mediate that relationship. The findings of the mediation effect of privacy and security and customer satisfaction are a unique contribution to the blockchain literature in the field of the hospitality industry. Hoteliers are encouraged to employ appropriate blockchain mobile payment services for better quality customer service and ensured safety and security, and in turn, loyalty intention.

**Keywords:** blockchain mobile payment service; service quality; privacy and security; customer satisfaction; loyalty intention

## 1. Introduction

Developing customers' loyalty intentions has become a vital issue in today's business including highly competitive hospitality industry because it functions as a fundamental differentiator to gain a competitive advantage (Al Karim et al. 2022, 2023; Pérez-Sánchez et al. 2021; Kandampully et al. 2015). Therefore, hoteliers must now, more than ever, focus on customer loyalty intentions, comprising the intention to use, loyalty, and positive WOM, and revisit intention by fulfilling their needs (Antoniadis et al. 2021; Prentice et al. 2020; Treiblmaier 2020). The widespread use of digitalization in the hospitality industry allows customers to search, book, and purchase travel online, which is different from conventional methods (Antoniadis et al. 2021; Khalifa 2018). However, customers sometimes get cheated and become worried about both their money and data security. Again, blockchain technology provides a wide range of phenomenal services, such as data transparency, robust security, immutability, accountability, lack of intermediary, efficiency, cost reduction, and reduced risk of data tampering, persuading the hospitality and tourism industry to implement this technology in a variety of areas (Marszałek and Szarzec 2022; Dogru et al. 2018; Irannezhad and Mahadevan 2020; Treiblmaier 2020). Thus, blockchain technology offers enormous potential to transform the way hospitality businesses are performing and increase customer loyalty and intention to use (Antoniadis et al. 2021; Dogru et al. 2018; Valeri and Baggio 2020). Consequently, several hospitality organizations in different countries, such as Australia, New Zealand, Germany, and Estonia, have implemented blockchain technology (Current Affairs 2018; Ma 2020; Marr 2018).

Nevertheless, in Bangladesh, blockchain technology implementation is already applied in the financial industry. For example, United Commercial Bank Limited, Prime Bank Limited, Standard Chartered Bank, and many other banks have launched blockchain

payment services to provide highly secure, faster, more accessible, and cost-effective transaction services (The Daily Star 2017; The Financial Express 2017). While this technology has grown in popularity and impacted the Bangladeshi financial industry, the hospitality industry has yet to recognize the potential of blockchain and its services.

The hospitality industry in Bangladesh is a significant sector of the economy (Rabiul et al. 2022; World Travel and Tourism Council 2020). This industry contributed 3% of the country's GDP (USD 9113.2 million) and 2.9% of total employment in 2019 (Rabiul et al. 2021; World Travel and Tourism Council 2020). This may locate many hospitality businesses in an unfavorable market position, particularly given the industry's major unwillingness to innovate and efficiently adapt to external market technologies (Filimonau and Naumova 2020). Therefore, for economic viability and development, it is the ideal moment and opportunity for hospitality businesses to use blockchain technology to enhance service quality, customer experience, and profitability and attain customers' loyalty intentions. However, several internet-based hotels and travel booking services providing sites in Bangladesh, such as ShareTrip, Stay, Ticketshala.com, and Winrooms.com, are making the travel experience more accessible for customers.

On the other hand, blockchain technology is a recent Fintech innovation that is still evolving swiftly (Sciarelli et al. 2022; Lou and Li 2017). Blockchain payment services are pioneering developments in the bargain that potentially disrupt tourism and hospitality firms (Nuryyev et al. 2020; Kwok and Koh 2019; Filimonau and Naumova 2020). The use of blockchain technology in the tourism and hospitality industry is being studied (Kwok and Koh 2019; Calvaresi et al. 2019), but there has not been enough empirical research into blockchain payments services, which may become a conventional payment method in the future (Karim et al. 2022; Nuryyev et al. 2020; Rad et al. 2018). Owing to the fast development of blockchain technology, it is vital to understand the elements that influence customers' loyalty intentions about blockchain payment services.

Therefore, due to inadequate research and knowledge, the hospitality industry of Bangladesh is experiencing the gap in blockchain technology implications. Hence, to fulfill the gap, the purpose of this study is to investigate the effects of blockchain payment services on customer loyalty intention through the mediating role of service quality, privacy and security, and customer satisfaction, which is yet to be explored in the Bangladesh hotel industry.

The contributions of this study are numerous. First, since blockchain is still a relatively new idea for Bangladeshi enterprises, this research can help boost the appeal of blockchain-based services by outlining many advantages that the Bangladeshi hospitality industry can gain by utilizing blockchain technology. Second, by investigating the mediating role of service quality, privacy and security, and customer satisfaction between blockchain-based services and customer loyalty relationships, this study provides how customer loyalty can be enhanced. This research will assist managers, administrators, and customers in comprehending the benefits of blockchain services, including hotel organizations, in implementing such technology-based services to build loyal customers.

## 2. Theoretical Framework and Hypothesis Development

### 2.1. Theory of Planned Behavior (TPB) and Technology Acceptance Model (TAM)

This study uses the theory of planned behavior (TPB) (Ajzen 1991) and the technology acceptance model (TAM) (Davis 1989) to examine the impact of blockchain payment services on customer loyalty intention in the hospitality industry. Since TAM and TPB have been employed in numerous studies to forecast and comprehend user perceptions of system use and the likelihood of adopting an online system, they are the most suitable tools for comprehending customer loyalty intention toward blockchain-based payment services (Lee 2009). In order to give a more complete model of blockchain-based online payment system evaluation and acceptance, this study recommends integrating the four characteristics of blockchain-based payment services with the TAM and TPB.

TPB contends that a person's loyalty intent, which is impacted by attitude, subjective norms, and perceived loyalty controls, collectively determines how they behave while carrying out certain actions (Ajzen 1991). The level of one's willingness to put in effort when engaging in particular actions is gauged by their loyalty intention (Ajzen 1991; Lee 2009). Simultaneously, according to the TAM's theoretical ground, perceived usefulness and perceived ease of use are the two critical determinants of loyalty intention to embrace modern technology (Davis 1989). Since its inception, numerous academics have merged TAM with extraneous elements and discovered it useful to correctly estimate technology adoption across a wide range of technologies and diverse environments, including information, software applications, and e-commerce (Sciarelli et al. 2022; Nuryyev et al. 2020; Kwok and Koh 2019; Kim and Woo 2016; Venkatesh et al. 2012; Venkatesh 2000). Due to its strong theoretical underpinning, it serves as one of the cornerstone theories for this research (Sciarelli et al. 2022; Nuryyev et al. 2020; Lou and Li 2017).

Numerous studies used the TPB and TAM to examine the factors influencing customers' loyalty intention (Lee 2009; Irawan et al. 2022). As the TPB model only addresses the influence of user-related or internal variables on loyalty intention, the TAM model might incorporate external elements, such as the effect of blockchain-based payment facilities on customer loyalty intention. By including the TAM and TPB models together, it becomes possible to examine the internal and external factors influencing customer loyalty intention.

### 2.2. Blockchain Mobile Payment Service and Its Features

Blockchain is viewed as a rigid database with a recurrent chain of blocks containing solitary data transactions that are transmitted among the users of that particular network using a decentralized mechanism (Rekha and Resmi 2021; Peters and Panayi 2016). Blockchain technology introduces a wide range of sophisticated features to the industrial and commercial worlds, facilitating development, scalability, and security, and easing many existing industrial and business operations (Al-Jaroodi and Mohamed 2019; Lee et al. 2019; Guo and Liang 2016). As a result, multiple industries, notably healthcare, finance, logistics, and hospitality, are also influenced by these new business models (Al-Jaroodi and Mohamed 2019; Önder and Treiblmaier 2018; Treiblmaier 2020). Organizations seamlessly provide blockchain-based services such as strong security, time-efficiency, and cost-effectiveness (Nowiński and Kozma 2017; Peters and Panayi 2016). Many hospitality and tourism organizations adopt blockchain payment services to enhance quality, customer satisfaction, and profitability. The following are the few specific features that represent blockchain payment services for the present study.

#### 2.2.1. Customer Familiarity

Customer familiarity is defined as the number of product-/service-related experiences a customer has gathered from previous interactions with a particular product or service (Olya et al. 2021). Customer familiarity includes customer experience, previous knowledge, and strength of inevitability (Olya et al. 2021) and affects customer behavior directly (Türkel et al. 2016). Familiarity with a product or service grows when people use it more frequently; hence, familiarity with a firm and its goods is associated with customer loyalty intention (Olya et al. 2021). Moreover, Miraz et al. (2020) found a strong correlation between customer familiarity and blockchain-based service uptake in the hotel business.

#### 2.2.2. Ease of Transaction

The payment system for the hospitality sector has been greatly impacted by the broad adoption of blockchain technology (Marszałek and Szarzec 2022; Önder and Treiblmaier 2018; Treiblmaier 2020). In their study, Kwok and Koh (2019) found that the blockchain system has the potential to ease cross-border transaction procedures and remove the necessity of currency conversion. A wide range of studies found that fundamental features of blockchain technology that ensure shorter technology-based transaction processes along with automating transaction services through smart contracts have a positive impact on

blockchain's ease of use (Lee et al. 2019; Lee 2019). Thus, it can be assumed that the ease of transaction procedure can positively impact blockchain-based services.

### 2.2.3. Transparency

Blockchain technology is a revolutionary tool that provides transparent and effective transactions to the users by merging cryptography, distributed systems, and consensus protocol authentically (Batubara et al. 2019; Kizildag et al. 2019; Maupin 2017). It is an incorruptible shared ledger of monetary transactions where transparency can be attained when recording and copying each transaction (Crosby et al. 2016; Golosova and Romanovs 2018; Gupta 2017; Willie 2019). A blockchain system, especially the public blockchain, enables its participants to access all the transaction data and information shared in the ledger and cannot be tampered with or erased by anyone (Golosova and Romanovs 2018; Kizildag et al. 2019).

### 2.2.4. Efficiency

According to several studies, participants in a blockchain system do not need mediators to build trust because all transactions and verifications are performed automatically in the network (Dogru et al. 2018; Gupta 2017). According to Willie (2019), hospitality organizations must evolve and become significantly more flexible in offering payment alternatives that guests select during transactions, to remain relevant and competitive. Though credit card transactions are more prevalent in the hospitality business than any other transaction, taking credit cards comes with the drawback of large commission fees that must be paid to credit card issuers (Willie 2019). Since blockchain technology eliminates the intermediate connection of the agent and eliminates commission fees or additional charges, the hospitality industry can help organizations reduce their operating and transaction costs for both business owners and customers (Nam et al. 2021; Treiblmaier 2019; Irannezhad and Mahadevan 2020; Willie 2019).

### 2.3. Blockchain Payment Services, Privacy and Security, Service Quality, Customer Satisfaction

One of the most remarkable features of blockchain is security solutions. A blockchain network does not require any central authority to approve or manage transactions (Irannezhad and Mahadevan 2020). Even in a blockchain system, to achieve consensus, every step of the transaction is approved and checked by all participants, increasing the level of transparency in the system (Dogru et al. 2018; Irannezhad and Mahadevan 2020; Rekha and Resmi 2021). Since these blockchains are stored on many computers and all transactions are rigorously guarded by unique cryptographic hashes, the hacking or changing the data history of the transaction by any individual in this system is impossible (Crosby et al. 2016; Dogru et al. 2018). By employing blockchain technology, the hospitality industry can introduce seamless transaction processes and improve financial management for both customers and service providers (Jobs 2021). Moreover, Willie (2019) suggests that blockchain can benefit the hospitality industry by making transactions easier, more secure, and transparent, improving business operations. So, the proposed hypothesis is as follows:

**Hypothesis 1 (H1).** *BPS positively affects privacy and security.*

Service quality has emerged as the most critical component in hospitality businesses' quest to gain a long-term competitive edge and hold existing customers (Ali et al. 2021; Lee et al. 2016). Service quality refers to customers' perceptions and assessments of service offerings by a business entity (Prentice et al. 2020). Ali et al. (2021) contended that services are imperceptible, so it is extensively complicated for the service providers to unwrap and customers to estimate. Several studies claim that responsiveness, safety and security, reliability, customization, and stability are all common sorts of service quality (Top and Ali 2021). Blockchain technology has a range of prosperous applications in the hospitality industry (Antoniadis et al. 2021; Dogru et al. 2018; Erceg et al. 2020; Valeri and Baggio

2020). By implementing this technology, hotel industries can ensure quality service, as the hospitality industry is extremely competitive, and the nature of blockchain-based services is remarkable (Dogru et al. 2018). According to Shahab and Allam (2020), businesses can reduce uncertainties, complexities, and the need for third-party involvement in transaction processes through blockchain payment services and improve the relationship between customers and service providers by enhancing service quality. So, based on the discussion above, the proposed hypothesis is offered:

**Hypothesis 2 (H2).** *BPS positively affects service quality.*

Top and Ali (2021) define customer satisfaction as whether the services or products provided meet the customer's expectations. Customer satisfaction has become the most important part of the hospitality industry, and compared to other industries, hospitality firms are assuring profitability and maintaining consumers through customer satisfaction (Jana and Chandra 2016). However, as hospitality firms become more competitive over time and customer preferences shift, meeting all the demands and needs of visitors has become difficult for these enterprises (Erceg et al. 2020; Preece and Easton 2019). The hospitality industry will undergo rapid transformation because of digitization, fueled by shifting customer preferences and needs for hotel and tourism-related products and services (Zsarnoczky 2018). Simultaneously, these new applications, such as blockchain payment, are affecting a variety of industries, including the hospitality industry, through their exceptional services (Treiblmaier 2019). According to Dogru et al. (2018), the immutable tracking feature of blockchain can increase tourists' efficiency, leading to customer satisfaction in the hospitality industry, as this technology has the potential to track a customer's overall travel details. Along with transaction convenience, blockchain technology can play a vital role in improving customer satisfaction in the tourism industry (Erceg et al. 2020). Thus, we propose:

**Hypothesis 3 (H3).** *BPS positively affects customer satisfaction.*

### 2.4. Privacy and Security, Service Quality, Customer Satisfaction, and Loyalty Intention

Privacy concerns are not unusual in the service industry. Personal information control, reproductive autonomy, accessibility to locations, confidentiality, and self-development are all aspects of privacy concerns (Sudigdo et al. 2019). Customers have become extremely concerned about the privacy and security of their private details on the wide global web, internet environment, and internet platforms and have demanded stronger protections (Rekha and Resmi 2021; Top and Ali 2021). Customers are typically unmoved by privacy when they sense their personal information is being employed only for the sake of the transaction. Customers become ever more worried about privacy if service providers use customers' personal data beyond the original transaction (Rekha and Resmi (2021). Customers are more willing to trust online payment service providers if they guarantee secure transactions and the secrecy of personal information (Shankar and Jebarajakirthy 2019). A transparent privacy and security strategy leads to better customer responses to the merchant (Orel and Kara 2014). Earlier studies have shown that privacy and security are positively significant to customer loyalty intention (Shankar and Jebarajakirthy 2019).

Moreover, the overall service quality evaluation leads to emotional reactions such as customer satisfaction (Makanyeza and Chikazhe 2017). Service quality has an extensive and supportive effect on consumer satisfaction and loyalty intentions (Makanyeza and Chikazhe 2017). This demonstrates that service quality is a crucial element that can generate customer satisfaction. Customers who leave due to dissatisfaction with the goods or service tend to tell others about their disappointment, which can influence other customers' loyalty intentions and, eventually, affect the service provider organization (Mokhtar et al. 2019).

Similarly, customer satisfaction has been identified as an evaluation that defines how delighted customers are with an organization's service offerings. Satisfaction has become

a significant concern for a company's product or service since it evaluates the level of expectation between the company's product and the customer's expectations (Ali et al. 2021). Customer satisfaction has a remarkable impact on both the company and the service they are providing, as customers happier with product performance and services means more sales and more profit (Top and Ali 2021). So, based on the discussions, the proposed hypotheses are:

**Hypothesis 4 (H4).** *Privacy and security positively affects customer loyalty intention.*

**Hypothesis 5 (H5).** *Service quality positively affects customer loyalty intention.*

**Hypothesis 6 (H6).** *Customer satisfaction positively affects customer loyalty intention.*

*2.5. Blockchain Payment Services and Customer Loyalty Intention*

Like other industries, the hospitality industry is continuously searching for new strategies to attain customer satisfaction and competitive advantage, increase productivity, and win customer fidelity (Willie 2019). Visitors consider loyalty intention in addition to motivation, purpose, and satisfaction when making decisions, and it is crucial to remember that a range of factors influences visitors when booking a hotel for travel purposes (Hu et al. 2020). The presence of customer loyalty intention (such as intent to use, loyalty, positive word of mouth, and revisit) is mandatory for the long-term sustainability of any hospitality firm due to the availability of a wide range of tourist spots and hotel businesses. Furthermore, market competition has become severe, so retaining loyal customers rather than acquiring new ones has become the major goal of these businesses (Pérez-Sánchez et al. 2021). Organizations can employ a range of blockchain-based payment services, such as easier transaction methods, transparency, efficiency, and immutability, to make the customer experience more satisfying (Antoniadis et al. 2021). Even the incredible feature of blockchain, smart contracts, can be used by the hospitality industry to facilitate transactions (Crosby et al. 2016; Dogru et al. 2018). Additionally, blockchain technology has been recognized as a powerful innovation competent to enhance the value of customer loyalty intention (Ebarefimia 2017). Pérez-Sánchez et al. (2021) explored that blockchain implementation in the tourism and hospitality sectors might positively impact customers' loyalty intention. The following hypothesis is offered based on these arguments:

**Hypothesis 7 (H7).** *BPS positively affects customer loyalty intention.*

*2.6. Mediating Role of Service Quality, Privacy and Security and Customer Satisfaction*

Along with cost-effectiveness, blockchain technology has the potential to assure secure transaction procedures as this advanced innovation is built on a system of cryptographic rules and strategies, such as hashing, consensus mechanisms, and timestamping for performing transactions (Papadopoulos 2015). By implementing this technology, businesses and customers could lessen the risk of credit card fraud, cybercrimes, identity theft, and personal data forgery (Hackius and Petersen 2017). Furthermore, firms must consider privacy and security concerns to maintain a long-term relationship with customers (Papaioannou et al. 2014). Likewise, many studies on privacy and security and customer loyalty intention found a positive association between these constructs (Shankar and Jebarajakirthy 2019).

What is more, according to Willie (2019), several hospitality companies have embraced blockchain technology to attain strategic and practical goals, namely, operational effectiveness, profitability, and efficiency. In addition, blockchain-based services are being employed in very few hospitality sector areas, aiming to enhance service quality. The findings of previous studies suggest that blockchain-based services can simplify business operations and accelerate service quality (Dogru et al. 2018; Rekha and Resmi 2021). With a favorable quality service, customers are likely to refer the product or service and spend more.

Moreover, service quality and customer loyalty intention are firmly related, and improved service quality is likely to improve loyalty intention (Myo et al. 2019). If the service quality is uneven or poorer than the requirement, customers may become unsatisfied and not engage again, leading to a significant loss in hotel revenues (Sudigdo et al. 2019). However, earlier studies discovered a direct and indirect connection between service quality and customer loyalty intention (Khalifa 2018; Sudigdo et al. 2019).

Erceg et al. (2020) and Dogru et al. (2018) specified that blockchain-based services might lead to guest satisfaction in the hospitality and tourism industry by tracking their current location, flight schedule, hotel check-in and out times, the location of their baggage, and transaction history, ensuring a cost-effective transaction process and streamlining the identity verification process. Customers are more likely to recommend the hotel's products, name, or services when they are more satisfied with the hotel's services (Khalifa and Mewad 2017). Further, satisfied customers will be loyal customers of the service provider corporation, and their aspirations to shift to another corporation will be restricted (Khalifa 2018). Numerous studies also stated that customer dissatisfaction could become a barrier to captivating customer loyalty intentions in the hotel industry that may negatively impact the business's profitability (El-Adly 2019; Kandampully et al. 2015). Following the discussions, the proposed hypotheses and the conceptual framework (Figure 1) are:

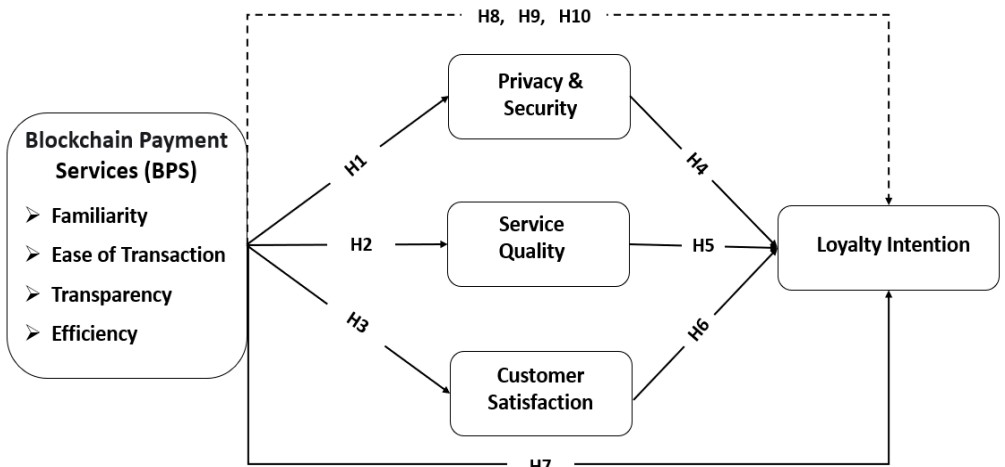

**Figure 1.** Conceptual Framework.

**Hypothesis 8 (H8).** *Privacy & security mediates the BPS and loyalty intention relationship.*

**Hypothesis 9 (H9).** *Service quality mediates the BPS and loyalty intention relationship.*

**Hypothesis 10 (H10).** *Customer satisfaction mediates the BPS and loyalty intention relationship.*

### 3. Methods

*3.1. Sample and Data Collection Procedure*

Although various sources indicated that only 44 star-rated hotels operate in Bangladesh (Karim and Rabiul 2022), the desired hotel list was not available on the government list or any other website. As a result, information was gathered using a structured questionnaire from guests at 4-star and 5-star hotels in the two most well-known cities, Chattogram and Cox's Bazar, in order to evaluate the hypothesis. Both cities contain a total of 22 hotels. Cox's Bazar is Bangladesh's most popular tourist attraction, and Chattogram is the largest port city. The managers of all 22 hotels were contacted to seek permission to conduct a survey on their customers, but only 17 granted permission to conduct a survey. As such, a total of 17 star-rated hotels were selected for data collection purposes from these two cities.

Moreover, permission was secured from the hotel's human resources and housekeeping managers to conduct the survey. We asked the housekeeping managers to hand out surveys in an open envelope to their hotel customers. Housekeeping departments gathered our questionnaires in sealed envelopes on our behalf.

Furthermore, before disseminating the questionnaires to the participants, the purpose of the study was clearly explained. The consent was obtained from all the participants verbally. The filled-out surveys were taken from the housekeeping departments after between six and eight weeks. The survey was started in February 2022 and ended in April 2022 after several follow-ups for questionnaire retrieval. A total of 600 questionnaires were circulated using the convenient sampling technique, which is a non-probability sampling method, and after deleting the outliers and missing responses, we received 326 usable responses, yielding a response rate of 54.33%.

*3.2. Measurements*

The blockchain mobile payment services (BPS) include four components, such as customer familiarity, ease of transaction, transparency, and efficiency, and were measured by 14 items adapted from an earlier study (Miraz et al. 2020; Nuryyev et al. 2020; Chen et al. 2022; Rekha and Resmi 2021). The items for service quality (five items) and privacy and security (four items) were adapted from Rekha and Resmi (2021), and customer satisfaction was measured by three items obtained from Chen et al. (2022). Lastly, the three items from Pérez-Sánchez et al. (2021) were used to measure loyalty intention. The items of the variables were in a five-point (1 = "Strongly Disagree" and 5 = "Strongly Agree").

# 4. Results

*4.1. Demographic Profile*

Table 1 demonstrates the demographic profile of the respondents. About 59.5% were men, and 40.5% were women. Regarding the age group, 52.5% belonged to the age group 26–35 years, 22.1% were between 18 and 25 years old, 21.8% were between 36 and 49 years old, and the rest were above 50 years old. Regarding educational qualifications, most of the participants (55.2%) had completed or were pursuing bachelor's degrees, and 34.7% of respondents were master's degree holders. Lastly, 34% of respondents were service holders, 27.6% were businesspersons, and 17.5% were students.

**Table 1.** Respondent profile (N = 326).

| Category | | Frequency | Percent (%) |
|---|---|---|---|
| Gender | Male | 194 | 59.5 |
| | Female | 132 | 40.5 |
| Age Group in years | 18–25 | 72 | 22.1 |
| | 26–35 | 171 | 52.5 |
| | 36–49 | 71 | 21.8 |
| | Above 50 | 12 | 3.7 |
| Educational Qualification | Bachelor's degree | 180 | 55.2 |
| | Master's degree | 113 | 34.7 |
| | Professional degree | 22 | 6.7 |
| | Other | 11 | 3.4 |
| Profession | Student | 57 | 17.5 |
| | Service holder | 111 | 34.0 |
| | Businessperson | 90 | 27.6 |
| | Professional | 18 | 5.5 |
| | Other | 50 | 15.3 |

*4.2. Common Method Bias and Multicollinearity*

Harman's single-factor test was run using SPSS 26 to examine the common method bias in the data. The total explanatory variance of a single component was 46.97%, below the cut-off value of 50% (Podsakoff and Organ 1986), and the total exploratory variance

of all factors was 68.12%, showing that common method bias was less likely to exist. This study examined the variance inflated factor (VIF) and its tolerance level to assess multicollinearity. According to Hair et al. (2014), a VIF value of greater than 3.0 confirms multicollinearity. Table 2 indicates that the VIF values were smaller than 3.0, indicating that multicollinearity was not a concern in this study.

**Table 2.** Mean, standard deviation, zero-order correlation and multicollinearity of the variables.

| Constructs | Mean | St. dev | 1 | 2 | 3 | 4 | VIF |
|---|---|---|---|---|---|---|---|
| 1. Blockchain payment services | 3.472 | 1.039 | 1 | | | | 2.703 |
| 2. Privacy and security | 3.870 | 0.888 | 0.698 ** | 1 | | | 2.420 |
| 3. Service Quality | 3.847 | 0.900 | 0.708 ** | 0.714 ** | 1 | | 2.600 |
| 4. Customer Satisfaction | 3.929 | 0.836 | 0.659 ** | 0.561 ** | 0.613 ** | 1 | 1.926 |

** Correlation is significant at the 0.01 level (2-tailed).

### 4.3. Measurement Model Assessment

Factor loading, composite reliability (CR), and average variance extracted (AVE) values are used to assess the measurement model's indicator reliability and convergent and discriminant validity (Henseler et al. 2016). Hair et al. (2020) suggested the value of individual items' loading to be >0.70, or in some points > 0.5 if the CR and AVE are above the cut-off value (CR > 0.7 and AVE > 0.5). Table 3 demonstrates the good internal consistency of the constructs, with loadings of >0.50, AVE > 0.50, and CR > 0.70.

**Table 3.** Construct reliability and validity.

| Construct and Items | FL | $\alpha$ | CR | AVE |
|---|---|---|---|---|
| Blockchain mobile payment services (BPS) | | | | |
| CF1: BPS is known to me | 0.550 | | | |
| CF2: BPS is fascinating to me | 0.526 | | | |
| CF3: I am aware of BPS | 0.633 | | | |
| CF4: BPS drew my interest | 0.696 | | | |
| ET1: BPS is clear and understandable | 0.795 | | | |
| ET2: BPS technology is easy to use | 0.746 | | | |
| ET3: Easy for me to become skillful at using BPS | 0.650 | 0.927 | 0.936 | 0.516 |
| TR1: BPS processes are transparent to me | 0.763 | | | |
| TR2: BPS provides in-depth access to transactions | 0.774 | | | |
| TR3: Applications of BPS are well described to me | 0.722 | | | |
| TR4: Usability of BPS is clear to me | 0.800 | | | |
| EF1: BPS technology helps make payments more effectively. | 0.769 | | | |
| EF2: Using BPS would enable to make payment more quickly | 0.793 | | | |
| EF3: BPS is a useful payment system. | 0.766 | | | |
| Privacy and security (PS) | | | | |
| PS1: BPS ensures authorized updating of shared information. | 0.866 | | | |
| PS2: BPS confirms lower risk of tampering of history of transactions. | 0.862 | 0.861 | 0.906 | 0.707 |
| PS3: BPS guarantees authorized information access. | 0.806 | | | |
| PS4: BPS confirms high level of accountability about transactions | 0.827 | | | |
| Service quality (SQ) | | | | |
| SQ1: BPS provides services as promised | 0.786 | | | |
| SQ2: BPS delivers prompt services to the customers | 0.827 | | | |
| SQ3: BPS instils confidence in the customers | 0.872 | 0.872 | 0.907 | 0.661 |
| SQ4: BPS provides best suitable services to the customers | 0.791 | | | |
| SQ5: BPS technologically up to date | 0.787 | | | |

**Table 3.** *Cont.*

| Construct and Items | FL | α | CR | AVE |
|---|---|---|---|---|
| Customer satisfaction (CS) | | | | |
| CS1: I am satisfied with the usage of blockchain-based payment service | 0.834 | | | |
| CS2: I am delighted to use blockchain-based payment service | 0.845 | 0.827 | 0.896 | 0.743 |
| CS3: My experience related to blockchain-based payment service is satisfying | 0.905 | | | |
| Loyalty intention (LI) | | | | |
| LI1: I will use blockchain payment service in my future travels | 0.917 | | | |
| LI2: I will use blockchain payment service for booking hotels for my future travels | 0.882 | 0.879 | 0.925 | 0.805 |
| LI3: I will encourage my friends, family and peers to use blockchain payment service | 0.891 | | | |

Note: FL = Factor Loadings; α = Cronbach's Alpha; CR = Composite Reliability; AVE = Average Variance Extracted.

The heterotrait and monotrait (HTMT) ratio was examined to confirm the discriminant validity. Table 4 suggests all constructs met the discriminant validity requirement, which was smaller than 0.85 (Henseler et al. 2016).

**Table 4.** Discriminant validity (HTMT Ratio).

| | 1 | 2 | 3 | 4 | 5 |
|---|---|---|---|---|---|
| 1. Blockchain mobile payment services | —— | | | | |
| 2. Loyalty intention | 0.725 | —— | | | |
| 3. Customer Satisfaction | 0.769 | 0.636 | —— | | |
| 4. Privacy and security | 0.792 | 0.662 | 0.665 | —— | |
| 5. Service Quality | 0.799 | 0.626 | 0.723 | 0.825 | —— |

*4.4. Structural Model Assessment*

The study ran PLS bootstrapping with 5.000 bootstrap samples (Hair et al. 2020). Table 5 indicates the t-values and *p*-values of the result. For instance, privacy and security ($\beta = 0.177$, $t = 0.078$, $p < 0.05$) and customer satisfaction ($\beta = 0.145$, $t = 0.081$, $p < 0.05$) were found to have a positive and significant impact on customer loyalty intention, while service quality ($\beta = 0.036$, $t = 0.448$, $p > 0.05$) had no significant impact on loyalty intention.

**Table 5.** Summary of hypotheses results (direct and indirect relationships).

| Hypothesis Path | β | t-Values | *p*-Values | Decision | $R^2$ | $f^2$ | $Q^2$ |
|---|---|---|---|---|---|---|---|
| Direct effects | | | | | | | |
| H1: BPS → PS | 0.723 | 25.767 | 0.000 | Supported | 0.522 | 0.105 | 0.365 |
| H2: BPS → SQ | 0.736 | 23.474 | 0.000 | Supported | 0.541 | 0.024 | 0.353 |
| H3: BPS → CS | 0.698 | 23.078 | 0.000 | Supported | 0.487 | 0.022 | 0.358 |
| H4: PS → LI | 0.171 | 2.219 | 0.013 | Supported | 0.479 | 0.001 | 0.373 |
| H5: SQ → LI | 0.036 | 0.448 | 0.327 | Not Supported | | | |
| H6: CS → LI | 0.138 | 1.694 | 0.045 | Supported | | | |
| H7: BPS → LI | 0.424 | 5.358 | 0.000 | Supported | | | |
| Mediating effects | | | | | | | |
| H8: BPS → PS → LI | 0.124 | 2.159 | 0.015 | Supported | Partial Mediation | | |
| H9: BPS→ SQ → LI | 0.027 | 0.445 | 0.328 | Not Supported | No Mediation | | |
| H10: BPS→CS → LI | 0.096 | 1.661 | 0.048 | Supported | Partial Mediation | | |

Notes: BPS = blockchain mobile payment service, PS = Privacy and security, SQ = Service Quality, CS = Customer Satisfaction, LI = Loyalty Intention, $R^2$ = Coefficient of determination; $f^2$ = effect size; $Q^2$ = Predictive Relevance.

The results from Table 5 also indicate that service quality could not mediate the relationship between blockchain payment services and loyalty intention ($\beta = 0.029$, $t = 0.479$, $p > 0.05$). However, customer satisfaction ($\beta = 0.100$, $t = 1.760$, $p < 0.05$) and privacy and security ($\beta = 0.127$, $t = 2.209$, $p < 0.05$) both partially mediate the relationship between blockchain payment services and loyalty intention. When both direct and indirect effects are significant, partial mediation occurs (Nitzl et al. 2016).

### 4.5. Model Quality

Hair et al. (2020) suggested that the $R^2$ values of 0.19, 0.33 and 0.67 mean small, moderate, and large effects on the endogenous variable, respectively. Table 5 shows 52.2% of the total variance in privacy and security, 54% in service quality, 48.7% in customer satisfaction, and 47.9% variance in customer loyalty intention. If the endogenous construct's $Q^2$ value is greater than 0, it is deemed predictive (Hair et al. 2020). Moreover, BPS, privacy and security and CS hold $f^2$ values of 0.105, 0.024 and 0.022, indicating that these three variables have small effects on the dependent variable CL, while SQ has no effect on CL. Table 5 further illustrates that the model had good predictive relevance, since the $Q^2$ value of all the endogenous constructs was more than zero.

## 5. Discussion

Blockchain payment services (BPS) was found to have a significant positive relationship with privacy and security (H1), service quality (H2), and customer satisfaction (H3). This is not surprising because the chances of hacking or manipulating the data history of transactions or any other personal data by any attacker in this system are difficult. For example, blockchain-based services are conserved on multiple computers, and all transactions are firmly secured by unique cryptographic hashes (Dogru et al. 2018; Crosby et al. 2016). In a range of service industries, including hospitality, customers are continually swamped with mundane product and service offerings. However, in the era of technology, they are more fascinated by technology-based creative solutions. The existing research studies of Erceg et al. (2020) and Dogru et al. (2018) found that customers in the hospitality industry were notably satisfied with the spectacular services provided by blockchain technology.

The result further revealed that privacy and security (H4) and customer satisfaction (H6) had a statistically significant association with customers' loyalty intention, but service quality (H5) had no impact on loyalty intention. The services of this technology, such as high security, reliability, transparency, immutability, and efficiency, have achieved enormous popularity within the hospitality industry and are crucial factors for attracting customer loyalty intention (Antoniadis et al. 2021). Moreover, this mobile-based application smoothly provides a wide range of services, such as booking and paying for hotels and travel tickets and redeeming reward points or vouchers to its customers, thus enhancing the loyalty intention of the customer through their satisfaction with this technology (Hu et al. 2020; Pérez-Sánchez et al. 2021). In contrast, Bangladeshi hospitality industry customers prefer to pay for hotel rooms or trip tickets using cash, credit card, or digital mobile banking services, which involve additional costs. Furthermore, cross-border money transfers also take a long time and are currently not supported by these digital mobile banking services. These are the few reservations about the insignificant relationship between service quality and customer's loyalty intention.

Moreover, blockchain payment service (BPS) was found to be significantly and positively related to loyalty (H7), supporting Pérez-Sánchez et al. (2021) study that revealed that the implementation of blockchain-based service in the hotel industry would generate a higher level of customers' loyalty intention, such as loyalty. Within a blockchain network, transactions are enforced, recorded, and validated immediately, automatically, and cost-effectively (Dogru et al. 2018; Gupta 2017). Additionally, customers may find this payment method much simpler and easier as the execution of the transaction process through blockchain's smart contracts takes a shorter amount of time than other traditional or digital payment systems (Kwok and Koh 2019).

Privacy and security (H8) and customer satisfaction (H10) were found to be influential mediators between BPS and customer loyalty intention. Along with ensuring cost-effective and faster payment methods, blockchain technology can streamline the identity verification process, guest tracking, hotel room booking, transportation booking, flight booking, and transaction process, increasing efficiency and attracting customers towards these services, which ultimately leads to customer satisfaction along with loyalty intention in the hospitality industry (Dogru et al. 2018; Erceg et al. 2020; Treiblmaier 2019; Willie 2019).

By implementing blockchain payment services within Bangladesh's hospitality industry, hoteliers can ensure a high level of profitability by establishing high-quality services (Rekha and Resmi 2021).

However, service quality (H9) did not mediate between BPS and customers' loyalty intention. Despite the potential benefits of blockchain technology, it has yet to be evaluated in the Bangladesh hotel business. There is still a potential mismatch between expectations and actual blockchain effects because actual service quality can only be assessed after large-scale deployments. As a result, customers might be unaware of the true value of blockchain and are unable to evaluate the service quality offered by BPS.

### 5.1. Theoretical Implications

This study is theoretically significant in many ways. According to the researchers' knowledge and available information, this study is one of the few to investigate customer loyalty intentions toward blockchain payment services in the Bangladesh hospitality industry. While earlier studies were carried out in other industries, such as banking, this study varies from prior studies. Furthermore, those studies examined the benefits and drawbacks of blockchain payment systems to offer the most modern options for individuals and enterprises. Hence, this research is essential to the literature on blockchain already published worldwide, including in Bangladesh. Next, very few studies have examined the mediating role of privacy and security and customer satisfaction between blockchain payment services and customers' loyalty intentions in the hospitality industry. This study adds value to the hospitality literature by investigating hospitality customers from a developing Asian country such as Bangladesh.

Moreover, through an empirical investigation of both internal and external factors influencing loyalty intention, this study broadens the area of research on technology adoption by customers in the hospitality industry of Bangladesh. This study differs from previous ones. It combines the TPB and TAM models with additional characteristics, such as customer familiarity, ease of transaction, transparency, and efficiency, to assess customers' intentions to remain loyal while paying for hotels. The findings thus contribute to the knowledge of technology-based blockchain payment systems and customer loyalty intention and enrich the TPB and TAM literature. Finally, this study advances knowledge of the understudied phenomena of blockchain payment services integration into operations and management in the hospitality industry of Bangladesh.

### 5.2. Managerial Implications

The findings of this study offer critical insights into blockchain for hospitality professionals. Hoteliers can increase trust and customer loyalty by using blockchain technology because it offers privacy and security, providing the best solutions to financial and non-financial industries in Bangladesh. One of the perks of employing blockchain for businesses is that it can streamline business operations by eliminating unwanted conflicts between stakeholders. Since hospitality and related industries frequently handle massive amounts of data, traditional systems are subject to mistakes. Top executives in the hospitality industry can remain stress-free with the precision of their data, mitigating the risk of data tampering by establishing blockchain-based services.

Additionally, hoteliers need to use blockchain to digitalize and revitalize transactions in a cost-effective way that may help them increase customer satisfaction and loyalty intention. In this way, hotels and other corporate entities can adopt blockchain technology to build customer-centric business activities and offer value-added services.

Moreover, practicing managers must not ignore the unique security solutions that blockchain provides and should implement this new technology to enhance the security and efficiency of their service offerings. Control over data is crucial in the hospitality industry, and during booking or check-in, customers need to provide a considerable amount of personal data, supported further by system resilience. So, blockchain technology can act as a facilitator that can ensure the immutability and reliability of the stored data and diminish

risk in certain situations, such as double spending, identity theft, and credit card forgery, ultimately benefiting firms and customers. Along with security solutions, hoteliers may establish blockchain-based loyalty programs that reduce costs, remove errors, enhance service effectiveness, and ensure customer satisfaction, all of which may lead to loyal customers (Ma 2020). In a nutshell, blockchain and its services can significantly alter the typical business operation process and work exceptionally well in customer retention.

Furthermore, without the government's assistance, it would be challenging for businesses and customers in Bangladesh to understand blockchain opportunities. Policymakers need to develop guidelines, initiatives, and regulations for hospitality businesses to embrace blockchain technology, and these policies will foster the firms' trust in blockchain innovation. Policymakers are also suggested to arrange awareness projects related to blockchain to provide insight and enlighten professionals, businesses, and their customers, thereby accelerating the acceptance of such innovative technologies (Sharma et al. 2021).

### 5.3. Study Limitations and Scope for Future Research

The limitations of this study are the following and may be used as the ground for future research. Primarily, the data were obtained from hospitality industry customers in Chattogram and Cox's Bazar. The sample size was 326, not covering the overall targeted population and perspective. However, collecting more data was impossible due to respondents' refusals and the government's restriction to travel to famous tourist destinations to control the spread of the COVID-19 virus. Future research studies can be conducted by gathering more data and increasing the sample size, which necessitates visiting more locations to understand the effects of blockchain payment services in the hospitality sector. Second, the respondents were selected using a convenience sampling technique with a low potential for portraying the entire population. Future studies may consider the probability sampling technique for a more generalized outcome. Third, the present study mainly focused on the perception of blockchain payment services towards loyalty intention in the hospitality sector, restricting the generalizability of the results in other industries. Fourth, this study only looked at four blockchain payment services (such as customer familiarity, ease of transaction, transparency, and efficiency), even though blockchain technology provides many other service benefits. Future research could include new blockchain services as financial and technological solutions for customers in the hospitality industry. Fifth, this study has used the TAM model as a theoretical framework, and this is a potential limitation of this study. So, future studies may use other technology-related theoretical frameworks, namely UTAUT and UTAUT2. Finally, future research may investigate other potential areas, such as the difficulties of attaining customers' loyalty intention in blockchain-based services, the challenges of deploying modern technology in a developing country such as Bangladesh, the complications of this technology, the customer experience and engagement with blockchain, and customer trust of blockchain-based services.

**Author Contributions:** Finding gaps, writing literature, data collection, analysis, and writing contribution, R.A.K., M.I. and S.K.; Establishing the problem statement, formulating the hypotheses, and improving the organization of the paper, M.K.R. and P.P. All authors have read and agreed to the published version of the manuscript.

**Funding:** There is no external funding for conducting this research.

**Institutional Review Board Statement:** Not applicable.

**Informed Consent Statement:** Not applicable.

**Data Availability Statement:** Data are available on demand.

**Conflicts of Interest:** The authors declare no conflict of interest.

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
