# Peer review of "Can Blockchain Payment Services Influence Customers’ Loyalty Intention in the Hospitality Industry? A Mediation Assessment"

_admsci, doi:10.3390/admsci13030085_

Round 1

Author Response

Dear Respected Reviewer:

Greetings! Please find the attachment of the author's response. We have tried to address all constructive and valuable comments. If there is any comment left, we will address it. Thank you for your support and guidance. Grateful. 

Reviewer 2 Report

I would like to thank the author(s) for the opportunity to read the manuscript. The study is well written and researched.  The paper is interesting, since Blockchain is a fruitful area of research. Be careful of some typos at the abstract and at the beginning of the manuscript.

Author Response

(The authors gave the same response as above.)

Reviewer 3 Report

1. In the Abstract you must correct the sentence "Customers' (N=326) opinion were a were... I would change the word Customers' with Respondents', there are also some English spelling mistakes.

2. In Chapter 2.2.2. The ease of transaction in line 131 (under& Treiblmaier, 2018) isn't written correctly.

3. In Chapter 4 .1. Demographic Profile. 

Why weren't any respondents above 66 years old? Write in the Article or in the Study limitations at the end of the Article.

4. Your total exploratory variance of a single component was 46.97%. What was the total exploratory variance of all factors?

5. In Table 3. you didn't write Cronbach alpha results for factors. Can you design the table differently so that α, CR, AVE can be seen better?

6. In table 5. is missing values q2 and f2.

7. You didn't explain the mediating effects in Chapter 4. Results. Explain it at the end of  4. Chapter.

8. Correct the References.  For example ( DOI: or https://doi..., No. or without No....)

Author Response

(The authors gave the same response as above.)

Reviewer 4 Report

The topic is really interesting as to its novelty and relevance. Furthermore, analyzing it in a developing scenario as it occurs with Bangladesh, represents a strength.

The theoretical framework is also correct and well developed. TAM as an evolution of TRA.

Data collection and data analysis are well presented. The paper counts with ten hypothesis with a clear TRA basis and perfectly justified.

All in all, I considered author have made a great work. I fully suggest its publication as it is.

Author Response

(The authors gave the same response as above.)

Round 2

Reviewer 1 Report

In my first review I wrote to the authors "There is a serious flaw and the authors should redo the model. All the marketing literature, at least that which is consistently and relevantly cited, considers satisfaction to be a construct that is the consequence of a purchase; satisfaction never precedes the purchase intention."
That is: either the authors change the designation of satisfaction or they change the designation of intention to use. In my view what the authors want to measure is not "usage intention". What they want to assess is "intention to continue to use" or something similar. "Intention to use" is assessing whether consumers will use, so it is assessing something that will occur in the future. "Intention to continue to use" measures something that has already occurred and hence the authors can measure satisfaction. In reality the intention to continue to use is close to the concept of loyalty. There is some confusion in the designations assumed by the authors. I mention once again. Satisfaction, Loyalty, Intention to use, are elementary constructs and much studied and therefore should be inserted in the model taking into consideration what is prescribed in referenced and solid articles.

Author Response

Please see attached file for the authors' response. 
